# Efficiently Substituting Dietary Fish Meal with Terrestrial Compound Protein Enhances Growth, Health, and Protein Synthesis in Largemouth Bass

**DOI:** 10.3390/ani14152196

**Published:** 2024-07-28

**Authors:** Fang Chen, Zhirong Ding, Zeliang Su, Junfeng Guan, Chao Xu, Shuqi Wang, Yuanyou Li, Dizhi Xie

**Affiliations:** 1College of Marine Sciences, South China Agricultural University, Guangzhou 510642, China; chenfang@scau.edu.cn (F.C.); 16620469962@163.com (Z.D.); zeliangsu@stu.scau.edu.cn (Z.S.); a374168380@163.com (J.G.); xuc1213@outlook.com (C.X.); yyli16@scau.edu.cn (Y.L.); 2Guangdong Provincial Key Laboratory of Marine Biotechnology, Shantou University, Shantou 515063, China; sqw@stu.edu.cn

**Keywords:** *Micropterus salmoides*, fishmeal substitution, growth, health, protein synthesis

## Abstract

**Simple Summary:**

Simple Summary: Fishmeal (FM), widely used in aquaculture, restrains the sustainable development of aquaculture due to its limited resource and high price. Numerous studies of FM substitution have mainly been conducted on terrestrial protein (plant and animal protein), indicating that inappropriate substitution of dietary FM can adversely affect the growth, health, and metabolism of carnivorous fish species. In this study, a terrestrial compound protein (Cpro) with chicken meal, bone meal, and black soldier fly protein were successfully developed and applied in the low-FM diets of largemouth bass. The results indicated that the dietary FM of largemouth bass could be effectively reduced to at least 18% by the Cpro, which is beneficial to health, digestion, and protein synthesis for maintaining growth.

**Abstract:**

Inappropriate substitution of dietary fishmeal (FM) can adversely affect the growth, health, and metabolism of carnivorous fish species. To effectively reduce the amount of dietary FM in carnivorous largemouth bass (*Micropterus salmoides*), a terrestrial compound protein (Cpro) with chicken meal, bone meal, and black soldier fly protein was used to formulate four isoproteic (52%) and isolipidic (12%) diets, namely T1 (36% FM), T2 (30% FM), T3 (24% FM), and T4 (18% FM), for feeding juveniles (initial weight: ~12 g) for 81 days. Results indicated that the growth performance, feed efficiency, and morphological indicators, as well as muscle texture and edible quality of fish, did not differ significantly among the four groups. However, the muscle protein contents and ATP/AMP ratio of fish in the T4 group were significantly increased in comparison with those of fish in the T1 group, while the opposite was true for muscle glycogen. Compared with the T1 group, high serum total amino acid and MDA contents, as well as low AST activities, were observed in the T3 and T4 groups, and relatively high intestinal trypsin and lipase activities were found in the T2–T4 groups. The transcripts of intestinal proinflammatory cytokines (*il-1β*, *il-6*, and *tnf-α*) were downregulated in the T2–T4 groups compared with T1 group, while the expression of anti-inflammatory cytokines (*il-10*) and tight junction (*zo-1* and *occludin*) showed the reverse trend. The mRNA expression of positive regulators related to protein synthesis (*sirt1*, *pgc1-α*, *pi3k*, and *akt*) were significantly upregulated in the muscle of fish fed diets T3 and T4, while their negative regulators (*4e-bp1*) mRNA levels were downregulated. The results indicate that the dietary FM of largemouth bass could be effectively reduced to at least 18% by the Cpro, which is beneficial to health, digestion, and protein synthesis for maintaining accelerated growth.

## 1. Introduction

Aquatic feed, particularly for the diets of carnivorous fish, heavily depends on fishmeal (FM) that have well-balanced nutrients compositions and good palatability. However, the extensive use of FM is not sustainable. As the demand for aquatic feed is estimated to increase by 37.4 million tons by 2025, alternative protein sources need to be explored [1]. Numerous studies of FM substitution have mainly been conducted on terrestrial protein (plant and animal protein) as substitutions for FM [2,3]. However, replacing fish meal with a high proportion of animal protein often leads to an amino acid imbalance or deficiency in certain essential amino acids (EAAs) in the feed, resulting in the decreased growth and digestibility of farmed fish [4,5,6]. Additionally, long-term application of low fish meal feed can change the composition of fish intestinal microflora and thus impact gut health [7,8,9]. Consequently, dietary FM replacement efficiency has emerged as a pressing issue to be solved addressed in the practical implementation of aquaculture.

Largemouth bass (*Micropterus salmoides*) is a highly valued freshwater fish species in China, renowned for its rapid growth and significant commercial worth. Like other carnivorous fish, largemouth bass has high dietary supplementation requirements for protein (42~51.6%) and FM (commercial feeds containing up to 50% FM), which limit its sustainable development [10,11]. Thus, many efforts have been devoted to developing FM substitutions to minimize the dependence of fish diet on dietary FM. The inclusion of a single protein source, such as soybean meal, fermented soybean meal, or chicken meal, can serve as a viable substitute for FM in aquatic feed; however, it is still necessary to supplement with a minimum of 28.8% FM [11,12,13]. Both animal protein (imbalanced amino acid compositions) and plant protein (anti-nutritional factors) have some limitations compared with fish meal. Compound protein sources derived from terrestrial animal and plant protein can be used to compensate for the insufficient single protein source that is replaced by FM, which leads to a higher replacement rate of dietary FM, reducing its level to 16% [14,15,16,17].

Lately, we have developed a terrestrial compound protein with soybean meal products, poultry byproduct meal, and cottonseed protein concentrate. This innovative formulation effectively reduces the dietary FM level to 6% in golden pompano (*Trachinotus ovatus*) [18,19]. This terrestrial compound protein was also used to replace some of the dietary FM of largemouth bass, while 30% FM supplementation was still required to add in their diets [20]. The present study aimed to enhance the reduction of FM in largemouth bass formula feed by an optimizing terrestrial compound protein (Cpro) with chicken meal, bone meal, and black soldier fly protein. The growth performance, health, muscle quality, and protein metabolism of fish were analyzed after an 81-day feeding trial, for the sake of evaluating the feasibility and suitable replacement level of FM with the Cpro. These results will provide a valuable reference in the development of efficient and low FM formula feed for largemouth bass.

## 2. Materials and Methods

### 2.1. Design of Experimental Diets

The Cpro (57.01% crude protein) consisted of chicken meal, bone meal (obtained from porcine by-products), and black soldier fly protein with a proportion of 45:45:10, which was used to replace 17–50% dietary FM in the compound feed of largemouth bass. Four isoproteic (52%) and isolipidic (12%) diets with varying levels of fish meal (FM) were prepared, namely T1, T2, T3, and T4 containing 36%, 30%, 24%, and 18% FM, respectively. The preparation and storage methods of diets were conducted in accordance with our previous study [19]. The dietary formulation and nutritional compositions are presented in Table 1.

### 2.2. Fish and Culture Management

The juvenile largemouth bass utilized in this experiment were procured from a nearby fish hatchery (Foshan, China). The juveniles (initial weight: ~12 g) were domesticated in large floating cages (1.5 m × 1.5 m × 2 m) at the Zengcheng Teaching & Research Base of South China Agricultural University. During the one-week domestication, the fish were fed with a mixture of equal amounts of the four experimental diets; then, 560 fish were randomly distributed to 16 cages (1 m × 1 m × 1.5 m, 35 fish per cage) and cultivated for 81 days. For the feeding trial, the fish were fed with their respective diets to apparent satiation twice a day at 6:00 and 17:00, and the weight of the diet consumed during the whole trial period was recorded. The water temperature ranged from 30 °C to 35 °C, and the dissolved oxygen level remained above 5.0 mg/L.

### 2.3. Sampling and Evaluation of Growth Performance

After the completion of the culture experiment, a 24 h starvation was imposed on the fish. The number and weigh of fish in each cage were recorded to assess their growth performance, including parameters such as survival rate (SR), weight gain rate (WGR), specific growth rate (SGR), and feed conversion ratio efficiency (FCR). The body, liver, and viscera of six fish randomly selected from each cage were weighed to analyze their morphological characteristics. Growth performance, feed utilization, and morphological characteristics were calculated using standard formulas, as we previously reported [18,20].

Additionally, another six fish from each cage were randomly sampled to collect plasma, intestine, liver, and muscle samples for the evaluation of biochemical indicators and gene expression. Additionally, four fish randomly selected from each cage were used to determine the muscle texture properties and nutrient composition of the whole fish. The collected samples were quick-frozen in liquid nitrogen and preserved in a −80 °C freezer with the exception of samples used for analyzing muscle texture properties.

### 2.4. Proximate Composition Analysis of Whole Fish and Muscle

The proximate analysis was conducted following the method previously described [18,20]. In brief, moisture content was determined by subjecting samples to a drying process in an oven at a temperature of 105 °C for 4 h. The content of crude lipid was measured using the Soxhlet extraction method (ST 255, Soxtec, Foss, Hillerød, Denmark). The crude protein content was computed by determining the total nitrogen content with the Kjeldahl method (KDN-102C Nitrogen Determinator, Xianjian instrument Co., Ltd., Shanghai, China). The content of ash was determined by combusting samples in a muffle furnace at 550 °C for 5 h.

### 2.5. Analysis of Muscle Glycogen and Adenine Nucleotides

Muscles were weighed and homogenized with saline solution, and immediately centrifuged at 10,000 r/min for 10 min; then, the supernatants were collected for measuring the contents of glycogen, adenosine triphosphate (ATP), and adenosine monophosphate (AMP). The contents of muscle glycogen were determined using a commercial assay kit (A043-1-1, Nanjing Jiancheng Bioengineering Institute, Nanjing, China), and the contents of ATP and AMP were measured with commercial assay kits (J38518 and J38447, Giled Biotechnology, Wuhan, China).

### 2.6. Analysis of Muscle Quality

The muscle quality, including edibility and textural properties, was assessed following the methodology outlined in a previous study [18]. Specifically, to determine the cooking percentage (CP) and water holding capacity (WHC) of fresh muscle, the initial weight of the clean fresh muscle was recorded. Subsequently, the weight of the muscle after being cooked in 100 °C water for 5 min and blotted with absorbent paper was measured again. The CP was presented as the percentage of the initial and cooked muscle weights. To measure WHC, the fresh muscle was first drained of blood using bibulous paper and then weighed. Then, the flesh muscle was enclosed in filter paper and extruded by 1 kg standard weight for 5 min. The muscle was discreetly removed from the filter paper and weighed. For analysis the textural properties of muscle, target samples were cut into square shapes (width: 2 cm × 2 cm; thickness: 0.5 cm) and preserved at 4 °C for 24 h prior to being tested with the Texture Analyzer (Universal TA, Shanghai turnkey pull, Shanghai, China) at room temperature. The adhesiveness, hardness, elasticity, chewiness, cohesiveness, adhesion, and resilience of muscle were determined with a cylindrical probe (Φ 5 mm) using the following parameters: the pressing speed and pressing depth were set at 0.8 mm s^−1^ and 60% of the sample thickness, respectively. Each sample underwent two measurements with a time interval of 30 s between them. The tenderness of muscle was assessed by an oblique edge probe with a blade thickness of 0.5 mm s^−1^ and a cutter speed of 1 mm s^−1^. All measurements were conducted in triplicate.

### 2.7. Analysis of Biochemical Indicators and Enzyme Activities

Serum total protein (TP), total amino acids (TAA), globulin (GLOB), albumin (ALB), aspartate aminotransferase (AST), alanine transaminase (ALT), urea nitrogen (BUN), and blood ammonia (BA), as well as serum and intestine malondialdehyde (MDA), total antioxidant capacity (T-AOC), alkaline phosphatase (ALP), and acid phosphatase (ACP), were determined with commercial kits (Nanjing Jiancheng Bioengineering Institute, Nanjing, China) following the manufacturer’s guidance.

Additionally, the activities of intestinal lipase (LPS), amylase (AMS), and trypsin (TPS) were detected with the commercial kits from Nanjing Jianceng BioEngineering Institute, referring to the manufacturer’s instructions.

### 2.8. Quantitative Real Time PCR

The mRNA expression of genes pertinent to inflammation (*il-1β*, *il-6*, *il-10,* and *tnf-α*), tight junction (*tgf-β*, *occludin*, *zo-1*, and *claudin-3*), and energy sensing (*sirtl*, *pgc1-a*, *pi3k*, *akt, s6k1*, *mtor*, and *4e-bp1*) of largemouth bass were quantified by real-time PCR. The specific primers utilized in this work are presented in Appendix A. Trizol reagent (Takara Biomedical Technology Co., Ltd., Shiga, Japan) was utilized to extract total RNA from the samples; then, 1 mg of isolated RNA was reversely transcribed into cDNA utilizing the PrimeScript™ RT commerical kit (Takara Biomedical Technology Co., Ltd., Japan). The target genes were amplified using TaKaRa SYBR Premix Ex TaqII in the CFX96 Real-Time PCR Detection System (Bio-Rad Laboratories, Inc., Hercules, CA, USA) following the manufacturer’s guidance. Each sample was tested in three parallels, and the relative expressions of the target genes were calculated with the comparative 2^−ΔΔCt^ method by reference with β-actin. The use of only one housekeeping gene may affect the interpretation of the results.

### 2.9. Statistic Analysis

The data are expressed as mean ± standard error of the mean (SEM) and were analyzed through one-way analysis of variance (ANOVA) with SPSS software (Ver26.0, Chicago, IL, USA). The differences between groups were assessed with the Turkey Kramer method and considered statistically significantly at *p* < 0.05.

## 3. Results

### 3.1. Growth Performance and Proximate Composition of Whole Fish

The results of Table 2 indicate that no statistically significant differences were observed in IBW, FBW, WG, SGR, SR, FCR, HSI, VSI, and CF among the four groups (*p* > 0.05). Notably, fish in the T4 group exhibited numerically superior values in FBW, WG, and SGR in comparison with those in the T1 group. Furthermore, no obvious differences were found in the content of moisture, as well as crude protein and lipid composition of whole fish among the four groups (*p* > 0.05).

### 3.2. Proximate Composition, Glycogen, and Adenine Nucleotides of Muscle

No distinct difference in the content of moisture, crude lipid, and ash were observed among the four groups according to Figure 1A (*p* > 0.05). However, a significantly higher content of crude protein was observed in the T4 group in contrast with that in the T1 and T2 groups (Figure 1A, *p* < 0.05). Additionally, the muscle glycogen contents were significantly lower in the T2–T4 groups compared with those of the T1 group (Figure 1B, *p* < 0.05). No noticeable variations in the muscle ATP and AMP contents were detected among the four groups (*p* > 0.05), while significantly higher muscle ATP/AMP ratios were measured in the T4 group (Figure 1B, *p* < 0.05).

### 3.3. Edible Quality and Textural Properties of Muscle

As shown in the Figure 1C,D, no noticeable variations in hardness, tenderness, chewiness, gumminess, adhesiveness, springiness, resilience, and WHC were detected among the four dietary groups (*p* > 0.05). The muscle cohesiveness in the T2 group and CP in the T3 and T4 groups were significantly lower than those in the T1 group (*p* < 0.05).

### 3.4. Serum Protein Metabolites and Health Indicators

As it can be seen from Table 3, no obvious differences were observed in the levels of TP, GLOB, and ALB or the activities of ALT, ALP, ACP, and T-AOC among the four groups (*p* > 0.05). Higher contents of TAA, BUN, and BA, as well as lower activities of AST, were observed in the T2–T4 groups compared with those in the T1 group (*p* < 0.05). Additionally, the levels of serum MDA were significantly higher in the T1 and T2 groups than those in the T4 group (*p* < 0.05).

### 3.5. Intestinal Digestive Enzyme Activity and Health Indicators

As indicated in Figure 2 and Appendix A, there were no statistical difference observed in the activity of intestinal trypsin and amylase between the T2–T4 groups and the T1 group (*p* > 0.05). However, the activity of intestinal lipase was significantly higher in the T2–T4 groups in contrast with that in the T1 group (*p* < 0.05). The levels of MDA, as well as the activities of intestinal T-AOC and ACP in T2–T4 groups, did not demonstrate any significant difference compared with those in the T1 group (*p* > 0.05). Moreover, a significant increase was observed in the activity of intestinal ALP in the T4 group compared with that in the T1–T3 groups (*p* < 0.05).

### 3.6. Expressions of Genes Associated with Inflammation, Tight Junction, Energy, and Protein Metabolism

The relative mRNA expression levels of genes associated with intestinal inflammation and tight junctions are depicted in Figure 3. In comparison with T1 group, the expression levels of genes related to proinflammatory cytokines (*il-6*, *il-1β*, and *tnf-α*) were downregulated in the FM substitution groups (T2–T4) (Figure 3A). Conversely, the mRNA levels of genes associated with anti-inflammatory cytokines (*il-10*) exhibited an opposite trend in the T3 and T4 groups. The expression of *tgf-β* (an anti-inflammatory cytokine) displayed an upward trend and exhibited a distinct difference in the T4 group (*p* < 0.05). The expression levels of genes associated with intestinal tight junctions, specifically *zo-1*, were significantly greater in the T3 and T4 groups compared with those in the T1 group. The mRNA expression levels of genes relevant to energy sensing and protein metabolism are demonstrated in Figure 3B. No significant difference was observed in the expression levels of regulatory factors related to protein metabolism (*akt*, *pi3k*, *s6k1,* and *mtor*) among the four groups (*p* > 0.05). The mRNA expression of muscular *sirt1* and *pgc1-α* genes were significantly elevated in the T3 and T4 groups compared with those in the T1 group (*p* < 0.05). Additionally, compared to the T1 group, the mRNA expression level of *4e-bp1* (a negative regulator of protein synthesis metabolism) was significantly reduced in the T4 group, while a decreasing trend in expression was observed in the T2 and T3 groups; however, these differences were not statistically significant.

## 4. Discussion

The evolutionary food preferences of carnivorous fish place particularly high demands on dietary protein and amino acids (AA); thus, these cultured fish species tend to accept dietary FM, but not dietary plant protein [21]. The high-priced and resource-limited FM resources contributes to the largest cost of the aquatic feed of carnivorous fish, negatively affecting the production of these fish [1]. Therefore, many endeavors have been conducted to seek for suitable FM substitutions from plant-, terrestrial animal-, insect-, and single cell-derived protein [22]. For example, the cottonseed protein concentrate can replace up to 50% of the dietary fishmeal (control diets contained 40% fishmeal) without any adverse effects on the growth performance of largemouth bass (initial weight ~15 g, weight gain ~48 g for 8 weeks) [23]. Similarly, approximately 57% dietary FM for this fish could be replaced by chicken plasma powder, where the control diets contained 35% FM, and the fish achieved an approximate weight gain of 65 g [24]. Ren et al. revealed that the dietary FM of largemouth bass juveniles (initial weight ~10 g) could be lowered to 16% by the supplementation of soybean meal and poultry byproduct meal, and the fish received a weight gain of 53.3 g, while their FCR was 15.05% higher than that of the control group [17]. A compound protein consisting of cottonseed protein concentrate, poultry by-product meal, and soybean meal was found to be a viable substitute for 80% of the dietary FM in largemouth bass diets (control diets contained 40% fishmeal), which reduced feed costs without negatively affecting growth performance or the environment [25]. Additionally, the replacement of dietary FM with dried black soldier fly larvae meal (DBSFLM) up to a 64% level did not alter the growth performance, hepatic, and intestinal histomorphology of Japanese seabass (*Lateolabrax japonicus*), suggesting the potential to be a desirable protein alternative source in this fish [26]. The dietary FM levels were reduced to a minimum of 18% using Cpro in the present study, resulting in fish achieving a weight gain of 106.35 g, which suggests that the Cpro is beneficial to the growth and feed efficiency of this fish species compared with those above FM substitutions reported in previous studies [17,23,24,25]. The Cpro may further reduce the dietary FM level of largemouth bass, which needs further study.

Serum physiological and biochemical indexes are important indicators for detecting the health and metabolism status in an organism [27]. The T-AOC activity serves as a direct indicator of the stress resistance and free radical scavenging capacity in fish, while MDA levels are widely employed for assessing tissue damage caused by oxidative stress [28]. Comparable T-AOC activities were observed both in the serum and intestine of fish fed FM and FM substitution diets in the current study. Additionally, the groups with FM substitutions exhibited lower levels of malondialdehyde (MDA) in both the serum and intestine compared to the FM group. Consist with this, Cheng et al. concluded that high dietary FM substitution for largemouth bass is adverse to health, lowering the antioxidant ability and increasing inflammation [22]. The anomalous expression levels of pro-inflammatory and anti-inflammatory cytokines, akin to the contents of MDA, serve as a manifestation of tissue damage. In this study, the expression levels of intestinal pro-inflammatory (*il-6*, *il-1β,* and *tnf-α*), together with the anti-inflammatory genes (*il-10* and *tgf-β*) in the Cpro group, were downregulated and upregulated, respectively. The gene expressions of anti-inflammatory cytokines were similarly elevated in the appropriate FM substitutions, while they were significantly downregulated with increasing dietary FM substitutions [23,29], which suggests that appropriate FM substitutions have beneficial effects on the inflammatory response. Liver injury induced by stress and inflammatory was characterized by abnormalities in ALT and AST levels and may be associated with increased serum AST and ALT contents [28]. In this study, dietary Cpro inclusion decreased the activities of serum AST, suggesting the Cpro inclusion is beneficial to maintain the liver health of largemouth bass, which may be attributed to the ability of black soldier fly protein to enhance fish’s antioxidant capacity and promote liver health by reducing serum AST and ALT levels [30,31]. Additionally, the nonspecific immunity properties (intestine ALP activities) and gene expression of intestinal tight junction proteins (*zo-1*) were modified in theT4 groups, which suggests that dietary Cpro is beneficial for maintaining the intestinal health of fish fed low-FM diets. All these above results indicate that the dietary FM of largemouth bass can be reduced to 18% at least using Cpro, which has no negative effects on the intestinal health of largemouth bass.

Serum TAA, ammonia, and urea nitrogen are the main metabolites of protein metabolism, which is widely used to evaluate the efficiency of protein and nitrogen metabolism in animal [29]. In this study, the contents of serum protein, ammonia, and urea nitrogen did not show significant differences between the dietary FM group and high-FM substitution groups (T3 and T4), while the serum T-AA contents enhanced progressively with the increase of the dietary FM substitution level. Serum T-AA contents generally reflected the uptake of amino acids from the feed. Li et al. found that serum T-AA decreased significantly by highly replacing FM with plant compound protein (rapeseed meal: cottonseed meal =3: 2) in yellow catfish (*Peltobargus fulvidraco*) diet [32]. The content of serum T-AA decreased significantly after adding 5% cottonseed meal hydrolysate in the diet of blunt snout bream (*Megalobrama amblycephala*); a similar result was also observed in turbot (*Scophthalmus maximus*) [33]. The results imply that the Cpro does not display adverse effects on the digestion and absorption of protein and amino acids for largemouth bass, which was confirmed by the comparable intestine trypsin activities shared in the four groups. The target of rapamycin (mTOR) pathway, an evolutionarily conserved pathway, is an important energy and amino acid sensor, balancing between protein synthesis and degradation [34,35]. The downstream effectors, such as *4e-bp1* and *s6k1*, were phosphorylated by mTOR components, which increases overall protein synthesis capacity of the cell [34]. The study of largemouth bass and golden pompano showed that appropriate FM substitution levels can activate mTOR pathway, including the upregulation of *s6k1* and *tor* and the downregulation of *4e-bp1*, which is inactivated in the fish fed diets with high FM substitution [23,36,37]. Consistent with previous studies, with the increasing of dietary Cpro levels, the expression level of *4e-bp1* in muscle was significantly downregulated, and the mtor expression levels showed an upregulation trend in this study. Accordingly, PI3K/Akt pathway, the classical upstream target of mTOR [38], was also upregulated in fish fed dietary Cpro compared with that of fish fed FM diets, which suggests that muscle protein synthesis may be improved by FM substitutions through the activation of the classical PI3K/Akt/mTOR pathway. The results were confirmed by the increase of muscle protein contents in the fish fed low-FM diets with Cpro.

In addition, sufficient nutrients and energy are also a guarantee for mTOR activation and protein synthesis [39]. The ATP/AMP ratios are widely employed for assessing the cellular energy status. In this study, the muscle ATP/AMP ratios exhibited an increase with escalating levels of dietary Cpro and were significantly elevated in the high Cpro groups compared to those in the control group. The results suggests that high dietary Cpro levels might activate the mitochondrial energy production of largemouth bass, leading to the mTOR activation and protein synthesis. This was indirectly supported by the fact that high expression levels of genes related sirt1/pgc1-α pathway, an important regulator of mitochondrial energy metabolism [40], were detected in fish fed high-Cpro diets. Supportively, the muscle glycogen and lipid contents in the T1 group were higher than those in the Cpro groups, and the content of muscle protein had the opposite trend, indicating that the Cpro may promote the mTOR activation and protein synthesis by supporting sufficient energy [41].

Apart from the health and growth of farmed fish, the increasing emphasis on high-quality products has garnered significant attention within the aquaculture industry. Edible quality and texture are important indicators for evaluating the muscle quality, which is affected by fish species, growth stages, and dietary nutrition [31]. Thus, muscle quality has also been employed as an important evaluation indicator for evaluating the quality of aquatic feed and its ingredients [18,31]. Accumulating studies are about the effects of alternative protein sources, including terrestrial animals, plants, and microorganisms on texture quality of different fish species, which yields inconsistent results, either showing significantly increased or decreased muscle firmness, or no significant difference on the firmness of muscle [31]. No significant differences were observed in the edible quality and textural properties, including muscle WHC, hardness, adhesiveness, springiness, chewiness, cohesiveness, gumminess, resilience, and tenderness between the dietary Cpro groups and FM group in present study. The high muscle protein contents were detected in the fish fed with low-FM diets (diets T4) compared with the those of fish fed control diets. These findings suggest that the dietary inclusion of fishmeal for largemouth bass could be reduced to at least 18% by utilizing Cpro, maintaining muscle texture and enhancing protein content, thus enhancing the nutritional quality of farmed fish and the dietary intake of consumers.

## 5. Conclusions

In conclusion, the current study demonstrated that the inclusion of Cpro, comprising chicken meal, bone meal, and black soldier fly protein could effectively reduce the dietary FM of largemouth bass to 18% without compromising growth performance. Furthermore, the Cpro formulation in the low-FM diets exhibited a positive impact on the intestine health and digestion, muscle quality, and protein synthesis of this fish species. These findings provide valuable insights for the development of cost-effective and low-FM compound feed tailored to largemouth bass.

## Figures and Tables

**Figure 1 animals-14-02196-f001:**
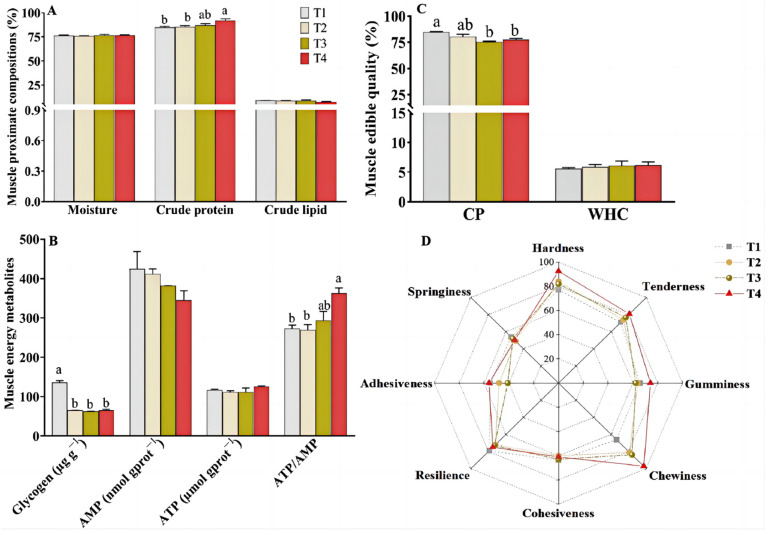
Muscle proximate compositions (**A**), energy metabolites (**B**), edible quality (**C**), and textural properties (**D**) of largemouth bass fed with different diets. Values are presented as the mean ± SEM (n = 4). Values in each row without sharing a common letter are significantly different (*p* < 0.05).

**Figure 2 animals-14-02196-f002:**
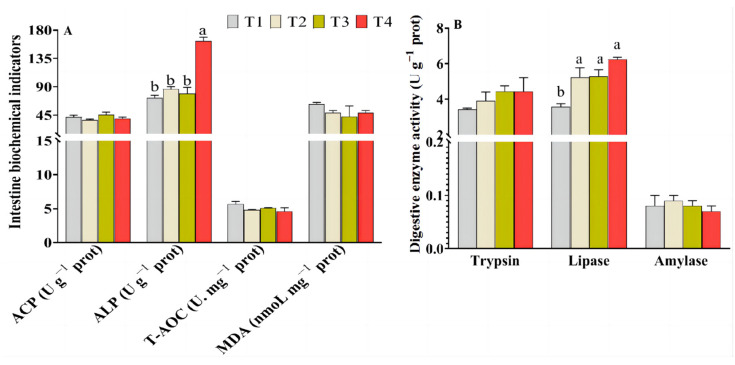
Health indicators and digestion of largemouth bass fed with different diets. (**A**) Intestine biochemical and antioxidant indicator; (**B**) intestine digestive enzyme activities. Values are presented as the mean ± SEM (n = 4). Values in each row without sharing a common letter are significantly different (*p* < 0.05).

**Figure 3 animals-14-02196-f003:**
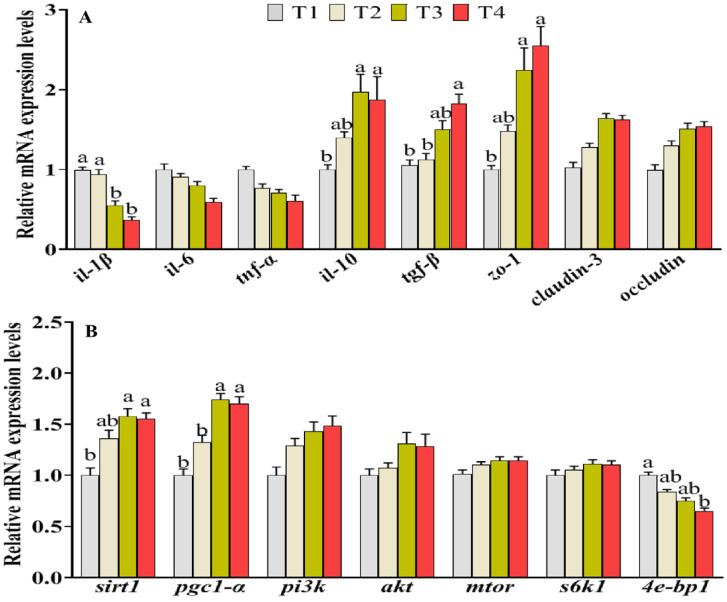
mRNA expression of genes associated with intestine inflammation and tight junction protein (**A**) and muscle protein metabolism (**B**) in largemouth bass fed with different diets. Values are presented as the mean ± SEM (n = 4). Values in each row without sharing a common letter are significantly different (*p* < 0.05).

**Table 1 animals-14-02196-t001:** Composition and nutrient levels of experimental diets.

Items	Dietary Groups			
	T1	T2	T3	T4
Fish meal	36.00	30.00	24.00	18.00
Cpro ^a^	0.00	8.50	17.00	26.00
Soybean protein concentrate	10	10	10	10
Cottonseed protein	10	10	10	10
Corn gluten meal	8	8	8	8
Fish oil	3.00	3.25	3.50	3.75
Soybean oil	4.88	4.63	4.38	4.13
Gluten flour	13.00	13.00	13.00	13.00
Premix compound ^b^	1.00	1.00	1.00	1.00
Calcium hydrogen phosphate	0.50	0.50	0.50	0.50
Choline chloride	0.50	0.50	0.50	0.50
DL-methionine	0.33	0.37	0.42	0.46
L-lysine	0.17	0.25	0.33	0.39
Spout corn bran	12.62	10	7.37	4.27
Proximate composition (%)				
Crude protein	52.79	53.96	50.37	50.62
Crude lipid	11.46	11.57	11.39	11.57
Crude ash	10.20	9.88	9.76	9.55
Dry matter	89.78	90.07	89.91	90.73

^a^ Cpro consisted of chicken meal, bone meal, and black soldier fly protein with a proportion of 45:45:10; ^b^ Premix compound: vitamin mixture (per kg mixture):VA: 1,100,000 IU; D3: 320,000 Iu; VB12: 8 mg; VK3: 1000 mg; VB1: 1500 mg; VB: 2800 mg; VC: 17 mg; VE: 8 mg; calcium pantothenate: 2000 mg; nicotinamide: 7800 mg; folic acid: 400 mg; inositol: 12,800 mg; VB6: 1000 mg. Mineral compound (per kg mixture):sodium fluoride: 2 mg; potassium iodide: 0.8 mg; cobalt chloride (1%): 50 mg; copper sulfate: 10 mg; calcium sulfate: 80 mg; zinc sulfate: 50 mg; manganese sulfate: 60 mg;magnesium sulfate: 1200 mg; sodium chlorde: 100 mg; zeolite powder: 15.45 g. All the dietary ingredients were provided by Yangjiang Haiyi Biotechnology Co., Ltd. (Guangzhou, China).

**Table 2 animals-14-02196-t002:** Growth performance and proximate composition of largemouth bass fed with different diets.

	Dietary Groups			
	T1	T2	T3	T4
IBW (g)	12.14 ± 0.00	12.13 ± 0.01	12.14 ± 0.00	12.15 ± 0.01
FBW (g)	110.33 ± 2.04	110.08 ± 3.07	117.31 ± 4.69	118.50 ±3.47
WGR (%)	809.01 ± 17.02	807.21 ± 25.69	865.98 ± 38.56	875.01 ± 28.21
SGR (% day^−1^)	2.72 ± 0.02	2.72 ± 0.04	2.80 ± 0.05	2.81 ± 0.04
FCR	0.77 ± 0.02	0.76 ± 0.04	0.73 ± 0.02	0.75 ± 0.02
SR (%)	98.33 ± 0.96	97.50 ± 1.60	98.33 ± 0.96	96.67 ± 1.36
VSI (%)	9.34 ± 0.45	10.08 ± 0.33	9.40 ± 0.46	10.15 ± 0.28
HSI (%)	2.46 ± 0.26	2.64 ± 0.30	2.83 ± 0.34	2.4 ± 0.15
CF (g cm^−3^)	2.80 ± 0.05	2.75 ± 0.10	2.78 ± 0.08	2.85 ± 0.03
Proximate composition (% dry weight)				
Moisture	73.87 ± 0.54	73.60 ± 0.67	72.58 ± 0.54	73.10 ± 0.61
Crude protein	68.26 ± 0.46	68.76 ± 1.01	67.88 ± 2.31	65.98 ± 0.91
Crude lipid	19.56 ± 1.09	19.77 ± 1.21	19.19 ± 1.59	19.25 ± 0.90
Ash	15.59 ± 0.58 ^a^	12.67± 0.31 ^b^	14.10 ± 0.00 ^ab^	15.16 ± 0.27 ^a^

Note: The data are presented as the mean ± SEM (n = 4). Values in each row without sharing a common letter are significantly different (*p* < 0.05). IBW: initial body weight (g). FBW: final body weight (g). Weight gain rate (WGR) = 100 × [FBW (g) − IBW (g)]/IBW (g). Specific growth rate (SGR, %/day) = 100 × ln [FBW (g) − IBW (g)]/days. Feed conversion ratio (FCR) = feed consumed (g)/[FBW (g) − IBW (g)]. Survival rate (SR, %) = 100 × (final number of fish)/(initial number of fish). Viscerosomatic index (VSI, %) = 100 × viscera wet weight (g)/FBW (g). Hepatosomatic index (HSI, %) = 100 × liver wet weight (g)/FBW (g). Condition factor (CF, g cm^−3^) = 100 × final body weight (g)/body length (cm)^3^.

**Table 3 animals-14-02196-t003:** Serum biochemical indicators of largemouth bass fed with different diets.

Items	Groups			
	T1	T2	T3	T4
Biochemical indicators				
TP (mg mL^−1^)	45.63 ± 3.49	45.12 ± 0.89	46.09 ± 2.60	58.49 ± 7.45
ALB (mg mL^−1^)	6.24 ± 0.45	8.10 ± 0.40	6.49 ± 0.84	7.04 ± 0.65
GLOB (mg mL^−1^)	41.55 ± 3.20	41.75 ± 1.88	41.20 ± 2.04	50.86 ± 6.64
TAA (umol mL^−1^)	56.62 ± 3.21 ^c^	83.46 ± 1.78 ^b^	105.88 ± 11.41 ^a^	111.91 ± 4.10 ^a^
BA (umol L^−1^)	338.33 ± 12.22 ^b^	784.26 ± 77.20 ^a^	373.98 ± 52.42 ^b^	463.26 ± 30.54 ^b^
BUN (mmol L^−1^)	233.79 ± 17.20 ^b^	439.57 ± 37.97 ^a^	220.14 ± 11.43 ^b^	268.47 ± 6.99 ^b^
ALT (U L^−1^)	2.72 ± 0.89	1.10 ± 0.27	2.03 ± 0.69	2.26 ± 1.09
AST (U L^−1^)	16.28 ± 0.34 ^a^	8.68 ± 0.41 ^b^	10.31 ± 1.51 ^b^	11.48 ± 0.79 ^b^
ACP (U L^−1^)	13.57 ± 0.69	13.64 ± 0.80	14.54 ± 1.38	15.77 ± 1.40
ALP (U L^−1^)	13.70 ± 1.01	14.57 ± 1.26	14.36 ± 3.86	18.80 ± 1.53
Antioxidant parameters				
T-AOC (U mL^−1^ prot)	0.30 ± 0.01	0.31 ± 0.02	0.32 ± 0.03	0.36 ± 0.01
MDA (nmoL mL^−1^ prot)	35.23 ± 2.31 ^a^	39.31 ± 3.91 ^a^	33.28 ± 0.82 ^ab^	29.00 ± 2.49 ^b^

Note: Values are presented as the mean ± SEM (n = 4). Serum total protein (TP), albumin (ALB), globulin (GLOB), total amino acids (TAA), blood ammonia (BA), urea nitrogen (BUN), alanine transaminase (ALT), aspartate aminotransferase (AST), acid phosphatase (ACP), alkaline phosphatase (ALP), total antioxidant capacity (T-AOC) and malondialdehyde (MDA). Values in each row without sharing a common letter are significantly different (*p* < 0.05).

## Data Availability

All data are contained within the article.

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
