# Peer review of "Efficiently Substituting Dietary Fish Meal with Terrestrial Compound Protein Enhances Growth, Health, and Protein Synthesis in Largemouth Bass"

_animals, 2024, doi:10.3390/ani14152196_

Round 1

Reviewer 1 Report

Comments and Suggestions for Authors

This manuscript entitled “Efficiently substituting dietary fish meal with terrestrial compound protein enhances growth, health, and protein synthesis in largemouth bass ” tried to replace fishmeal with a terrestrial compound protein at a high level. The study is meaningful, and the results are reliable. However, some points have to be modified before publication.

Abstract

1. “accelerated growth” is inappropriate since all the groups showed similar growth performance.

Introduction

1. line 43 delete “sources”, and replace “amino acid” with “nutrients”.

2. line 52 microflora or microbiota

3.Table 1 a,b should be superscript.

4.The authors should introduce the protein sources, including their advantages and disadvantages. Why did you mix them as a terrestrial compound protein? According to their nutritional composition?

5.Line 69-line 74 the previous terrestrial compound study is not highly related to the present study.

Materials and Methods

Line 84 Why did you use a proportion of 45:45:10?

Line 110 mg/L

Line 170 “Nanjing Jianceng”?

Results

Line 114 survival rate (SR),

The gene names for mRNA should be italicized, such as line 173-175.

Line 175 M. salmoides should be italicized. Carefully check other places.

It is lacking of definition for many abbreviations, such as IBM, FBW, and others.

The IBM was higher than 12g, not ~11g.

Discussion

Line 274-line 291: These sentences contain basic information and should be included in the introduction.

Line 293 No significant difference in WG and FE, so it was no beneficial effect.

Why Cpro can replace this high level of FM? Potential explanations?

The authors should analyze the amino acid composition of Cpro and fishmeal, as you mentioned, amino acids might be the primary parameters for protein sources.

Line 314-315 Normally, replacing FM with non-fishmeal sources causes liver damage. I can’t believe the AST and ALT activities in the substituted groups were reduced. Please give reliable explanations.

Line 318-319 “no any negative effects on the fish health”. The health parameters you analyzed were limited, so you can’t come to such a conclusion.

Line 31 lacks a Latin name, as for other places.

Use the common name of fish solely. Check the whole manuscript.

The discussion is not deep enough. It would be best if you focused on why the combination of these three ingredients had such effects based on the nutritional components. Add more references.

Comments on the Quality of English Language

A native speaker should polish the language.

Reviewer 2 Report

Comments and Suggestions for Authors

This manuscript investigates the potential of a specific compound to replace fishmeal (FM) in the diet of largemouth bass. different physiological parameters have been investigated leading to the conclusion that the substituton up to  The research is well-founded, but some minor methodological adjustments and improvements in graph/figure presentation could strengthen the findings. For this reason I suggest minor revisions and my specific comments are directly in the text to improve clearness. Addittionally, I will suggest a major text revision maybe with the help of native speaker. The discussions are nice but some statements a little redundant. I suggest to reduce it and condense the text to enhance clarity. Specific edits/quastions are done in the pdf attached to the format. 

Comments on the Quality of English Language

English quality is quite good, it reads very smoothly but minor mistakes here and there (as highlighted in the pdf version attached)

Reviewer 3 Report

Comments and Suggestions for Authors

Overall comments

The current study focused on the effects of using a terrestrial compound protein (Cpro) to reduce the amount of dietary FM in carnivorous largemouth bass (Micropterus salmoides), The manuscript is well-written, the relevance of research is clearly stated, and results observed are promising, albeit needing to be better described.  There are minor orthographic and grammatical errors along the manuscript. 

Some examples follow:

  • In line 63, "Either" should be replaced with "Both”
  • In line 64, "has" should be replaced with "have”

Specific comments

Introduction:

  • In lines 65-67, the authors state that a higher FM replacement is achieved with compound protein. However, compared to the other feedstuffs mentioned, 30% of FM is still required with compound protein, whereas only 25% FM is needed with the other feedstuffs. Please clarify this paragraph

Methodology:

  • Was the bone meal used in Cpro also obtained from chicken?
  • Why was only one housekeeping gene evaluated? Current standards suggest including at least two to prevent bias in result interpretation.

Results section:

  • In line 193, it should be HSI instead of “HIS”.
  • In line 213, replace “significantly low” by “significantly lower”
  • In line 244, replace “notably” by “significantly”
  • In line 247, replace “notable increase” but “significant increase
  • In line 193, "HIS" should be "HSI."
  • In line 213, replace "significantly low" with "significantly lower."
  • In line 244, replace "notably" with "significantly."
  • In line 247, replace "notable increase" with "significant increase."
  • Avoid adjectival phrases in the description of results. For instance, in line 215, "obviously high" should be replaced with "higher" or "the highest."
  • The sentence "These findings suggested..." in lines 225-227 belongs in the discussion section.
  • The sentence in lines 233-234, "Additionally, the levels of serum MDA were markedly higher in the T2 group than in the other three groups," does not accurately describe the results. Figure 2 shows MDA levels in T2 are the highest but only statistically different from those of T4.
  • Why are the results from serum biochemical parameters separated into two figures? This makes reading the results harder. Either re-write the results to match their order in the different figures (Health marker and digestion in Figure 2; protein metabolism in Figure 3) or alter the figures, keeping all serum results in one figure (Figure 2) and all gene expression results in another (Figure 3).
  • In Figure 3, please include the same abbreviations used in the results description, particularly those in line 231 (TAA, BUN, and BA).
  • Serum ammonia and urea nitrogen are only significantly higher in T2 compared to T1. Please revise the sentence in lines 231-232 to match the results shown in Figure 3.
  • In line 254, only IL-1β expression was significantly downregulated, and only in T3-T4. Avoid describing results without statistical significance. To point out trends, attach the p-value of these results, and it should at least be close to 0.05.
  • Specify that IL-10 was only significantly upregulated in T3-T4.
  • In Figure 2D, TGF-β1 is clearly upregulated in T4 compared to T1-T2. Revise the sentence in lines 256-258 accordingly.
  • According to Figure 2D, only ZO-1 is significantly upregulated in T3-T4. Thoroughly revise this paragraph to reflect the results shown in the figure.
  • In lines 265-266, "Additionally, down-regulation of 4e-bp1, a negative regulator of protein synthesis, was observed in the T2-T4 groups (particularly in the T4 group) compared to T1": this downregulation is only statistically significant in T4.

Discussion:

  • Revise the sentence in lines 284-285 from “and the fish received ~65 g weight gain” to “and the fish achieved an approximate weight gain of 65 grams.”
  • Following the above example, revise the sentence in lines 287-288.
  • In lines 294-296, specify which results the authors are referring to. In the present work, Cpro does not appear to contain plant protein.
  • Revise the sentence in lines 310-312. What do the authors mean by appropriate FM substitutions?
  • The sentence in lines 316-318 does not reflect the results observed in Figures 2-3. Justify how the changes induced by Cpro are beneficial.
  • In lines 325-326, how was this correlation obtained? It should be stated in the Material & Methods section.

In line 330, include “and a” before “similar result.”

  • The sentence in lines 334-335 does not have a verb.
  • Change “which is conductive” to “leading.”
  • If protein synthesis is being induced and fish growth is not affected, what is the authors’ hypothesis about what is happening to this protein?
  • Regarding the differences observed in muscle CP, what do these results mean for the consumer?
  • Do the authors find that the results obtained in gene expression are related to the total protein and glycogen levels in the muscle?

Conclusions:

Conclusions are general but overall reflect the results. Based on the authors' results, at what percentage of Cpro inclusion are the most beneficial results in intestinal health or muscle quality observed?

Comments on the Quality of English Language

There were minor English mistakes found. They were pointed out in the comments above.

Round 2

Reviewer 1 Report

Comments and Suggestions for Authors

None

Author Response

Thanks for your previous suggestions, we have made revisions accordingly.

Reviewer 3 Report

Comments and Suggestions for Authors

In the second revision of the manuscript, I encourage the authors to review some minor aspects, before publication:

1) Author response:Thanks for your good suggestions, the deployment of multiple reference genes enhances the precision of qPCR outcomes, an aspect that was unfortunately omitted in the initial experimental design, given the absence of a strategic plan for their incorporation.Unfortunately, the feeding trial concluded in October 2022, and the tissue samples were not adequately preserved for re-detecting gene expression using multiple reference genes.  

- Please mention in the methodology section 2.8 that the use of only one housekeeping gene may affect the interpretation of the results.

2) Author repsonse: Thanks for your suggestion, revision has been made as “a down-regulation of 4e-bp1, a negative regulator of protein synthesis, was observed in the T2-T4 groups (particularly in the T4 group) compared to T1.”

-While a decrease is observed in the remaining treatments, a significant downregulation is only evident in T4. Please ensure this distinction is clear.

3) Author response: The previous studies indicated that appropriate FM substitutions (replaced 25% FM) is beneficial for mRNA expression levels of anti-inflammatory cytokines, and reducing inflammatory responses.

- Please include in the discussion section that up to 25% FM replacement has a beneficial effect on the inflammatory response.

4) Author response: Thanks for your suggestion, the conclusion was revised into “Additionally, the nonspecific immunity properties (intestine ALP activities) and gene expression of intestinal tight junction proteins (zo-1) were modified in the T4 group”.

-The abbreviation ALP has been changed to AKP, but in Figure 2, it remains unchanged. Please update the figure accordingly. Also, specify how the increase in AKP and the downregulation of zo-2 expression induced by T4 benefit largemouth bass's antioxidant performance and inflammatory response.

5) Author response: The stimulation of muscle protein synthesis does not correspond with the change of growth observed in the fish fed low FM diets with Cpro, which may be attributed to the following reasons:

1) Compared with the control group (T1), no significant difference was observed in the growth of the Cpro groups (T4); however, the latter group exhibited an 8.15% enhancement in growth. 2) The synthesis of body protein serves as a critical indicator of fish growth. However, fish growth, in terms of weight gain, is also intricately linked with other metabolic processes, such as lipid metabolism and bone metabolism. These aspects require further investigation and study.

-This is relevant infromation and should be included in the discussion.

6) In lines 391-392 “While, the high muscle contents were detected in the fish fed with low FM diets (diets T4) 391 compared with the those of fish fed control diets.”

The authors should review this sentence. What does “high muscle contents” mean? Do they mean a higher cooking portion was obtained?

7) Author response: Thanks for your suggestion, revision has been made in Conclusions section: “In conclusion, the current study demonstrates that the inclusion of Cpro, comprising chicken meal, bone meal, and black soldier fly protein, can effectively reduce the dietary FM of largemouth bass by at least 18% without compromising growth performance Furthermore, the Cpro formulation in the low FM diets exhibits a positive impact on the intestine health and digestion, muscle quality, and protein synthesis of this fish species. ....”

While, the optimal substitution of dietary FM with Cpro requires further investigation, given that the maximum replacement level examined in this study was 18%.

-Instead of “by at least”, it should read “to 18%”.

Comments on the Quality of English Language

The English language has been revised as advised.
